

# Psychopathological symptoms in soccer referees: the role of psychological inflexibility and perfectionism

Félix Arbinaga[1], Emilio Moreno San Pedro[1] and María-Isabel Mendoza-Sierra[2]

[1] Department of Clinical and Experimental Psychology, University of Huelva, Huelva, Spain
[2] Department of Social, Development and Educational Psychology, University of Huelva, Huelva, Spain

## ABSTRACT

**Background**. Refereeing is associated with a high prevalence of mental health issues. Mental health problems are quite common in sport and referees are no exception. In the case of referees, psychological disorders have been associated with a number of factors and lower league officials appear more likely to experience psychological distress compared to their higher league counterparts.

**Aim**. The aim is to analyze the relationship between psychological inflexibility, perfectionism, and psychopathological symptomatology in soccer officials.

**Method**. A cross-sectional, anonymous, online study was conducted. Psychopathological symptoms were assessed using the Symptom Assessment-45 Questionnaire; the Acceptance and Action Questionnaire was used to assess psychological inflexibility; Perfectionism was assessed using the Multidimensional Perfectionism Scale. Participants are 156 active main referee (96.8% male), of whom 63.5% are at amateur level and 36.5% at semi-professional/professional level.

**Results**. No significant differences were found between amateur and semi-professional/professional referees in psychological inflexibility and psychopathological symptoms, except for paranoid ideation, where amateurs scored higher. Significant differences were observed in the total scores of maladaptive perfectionism, particularly regarding external influences, with amateurs scoring higher. In adaptive perfectionism, significant differences were noted in both total scores and achievement expectations, with amateurs obtaining higher scores. Psychological inflexibility demonstrated a strong predictive capacity for psychopathological symptoms ($\beta = .716$). When maladaptive perfectionism was incorporated in the model, it significantly predicted 17.6% of the variance. Adaptive perfectionism, however, did not significantly predict psychopathological symptomatology.

**Conclusion**. The results suggest that psychological inflexibility and maladaptive perfectionism are good predictors of psychopathological symptoms and mental health in referees. The status of amateur or semi-professional officials does not differentiate them from professional referees in terms of psychological disorders, but it does in terms of perfectionism. With a view to the future, it is important to intervene on these constructs, which are modifiable and facilitate their well-being.

Corresponding author
Félix Arbinaga,
felix.arbinaga@dpsi.uhu.es

## INTRODUCTION

Mental health problems are highly prevalent in sports (*Gulliver et al., 2015*; *Kilic et al., 2021*; *Poucher et al., 2021*), and referees are no exception (*Arbinaga et al., 2019*; *Lima et al., 2022*). In referees, psychological disorders are associated with intrapersonal factors such as marital status (being single), younger age, limited refereeing experience, history of injuries, and performance concerns (*Carson et al., 2020*; *Lima et al., 2022*). Interpersonal factors, such as dissatisfaction with social support (*Gouttebarge et al., 2017*; *Kilic et al., 2018*) and organizational factors, including abusive and toxic environments and issues with arbitral structures, have also been identified (*Webb et al., 2021*). Moreover, female officials are at a higher risk of mental health problems (*Carson et al., 2020*; *Lima et al., 2022*; *Vela & Arbinaga, 2018*; *Webb et al., 2021*), while lower-league referees are more likely to experience psychological distress compared to their higher-league counterparts (*Lima et al., 2023*). Amateur officials face a greater number of negative behaviors (*Webb et al., 2020*) and have greater concerns about being assaulted, partly due to their closer proximity to the public and their younger age (*Cuskelly & Hoye, 2013*). Furthermore, they often have fewer resources to effectively perform their refereeing duties.

*Kilic et al. (2018)* conducted a study among soccer referees from eight European countries, finding that 5.9% reported distress, 11.8% reported anxiety/depression, 9.1% reported sleep disturbances and 16.5% reported adverse alcohol use. Additionally, *Gouttebarge et al. (2017)* examined the prevalence of common mental disorders among professional soccer officials from various European countries, reporting a one-season incidence of common psychological disorders symptoms: 10% for distress, 16% for anxiety/depression, 14% for sleep disturbance, 29% for eating disorders and 8% for problematic alcohol use. A higher number of severe injuries (*Arbinaga, 2025*) and a lower degree of satisfaction with social support have shown to be significantly related to the occurrence of common mental disorder symptoms, with odds ratios (OR) of 2.63 and 1.10, respectively (*Kilic et al., 2018*).

Perfectionism is a widely studied construct within sports (*Hill et al., 2019*). In athletes, perfectionism is related to performance and plays a prominent role in both functional and adaptive aspects (*Rice, Lopez & Richardson, 2013*; *Robazza et al., 2023*; *Taylor et al., 2016*). Adaptive perfectionism is associated with factors such as achievement expectations and organization. Conversely, maladaptive aspects are characterized by high external expectations (imposed by family and coaches), fear of making mistakes, and reflections on the quality of performance (*Appleton, Hall & Hill, 2011*; *Lizmore, Dunn & Causgrove-Dunn, 2017*; *Madigan, Stoeber & Passfield, 2017*).

It should be noted that perfectionism is associated with several psychopathologies (*Stoeber & Otto, 2006*). The relationship between perfectionism and psychopathological symptoms has been extensively studied across various sports (*Gulliver et al., 2015*; *Hill et al., 2015*; *Nixdorf, Frank & Beckmann, 2016*; *Schaal et al., 2011*). For example, international athletes recognize that while perfectionism can serve as a source of motivation, it may also be associated with intrapersonal difficulties, such as worry or insomnia, and interpersonal challenges such as tensions with colleagues or technical team (*Hill et al., 2015*). Therefore,

in the sports context, perfectionism has been associated with anxiety (*Koivula, Hassmén & Fallby, 2002*; *Schaal et al., 2011*; *Stoeber et al., 2007*), depression (*Gorczynski, Coyle & Gibson, 2017*; *Gulliver et al., 2015*; *Nixdorf, Frank & Beckmann, 2016*; *Schaal et al., 2011*; *Wolanin et al., 2016*), and stress (*Crocker et al., 2014*; *Flett & Hewitt, 2005*; *Hall, 2006*; *Schaal et al., 2011*; *Tashman, Tenenbaum & Eklund, 2010*). These relationships were primarily found with maladaptive perfectionism, although it was also noted that adaptive perfectionism could be also associated with distress (*Hill et al., 2008*). It can be argued that inflexible individuals lack strategies tailored to specific situations and tend to use the same strategies regardless of context (*Crosby et al., 2013*). Perfectionism is also associated with adverse outcomes, including stress, poor mental health, pain frequency/intensity, and fatigue (*Molnar et al., 2012*).

Additionally, perfectionism has traditionally been conceptualized as a vulnerability factor (*Sirois & Molnar, 2014*), characterized by cognitive rigidity and behavioral inflexibility (*Delor, Douilliez & Philippot, 2019*). More specifically, perfectionism is associated with impaired functioning and difficulties in achieving optimal health (*Molnar et al., 2012*), mediated by processes such as behavioral disengagement, denial, self-blame (*Quartana, Campbell & Edwards, 2009*), and experiential avoidance (*Bisgaier, 2019*).

Conceptualizing perfectionism as a contextual behavioural construct has allowed researchers to explore its relationship to psychological flexibility (*Gentili et al., 2019*). Psychological flexibility can be defined as the process of engaging with the present moment as a conscious human being, fully and without unnecessary defenses—being as one is, rather than as one perceives oneself to be—while persisting or modifying behaviors in alignment with freely chosen values (*Hayes, Strosahl & Wilson, 2014*). The psychological flexibility model fosters adaptive coping through the six components: acceptance, cognitive defusion, present-moment awareness, self-as-context, being in contact with one's values, and committed action (*Hayes et al., 2006*; *Wicksell, Olsson & Hayes, 2010*).

The opposite of psychological flexibility is psychological inflexibility. The latter refers to the rigid dominance of certain unhelpful private events over effective actions, long-term goals, useful thoughts, and emotions (*Bond et al., 2011*). The behavioral pattern characterizing individuals with high psychological inflexibility is experiential avoidance, which becomes a generalized and rigid pattern, devoid of actions driven by what is meaningful to the individual (*Hayes, 2015*). Three key processes fundamentally characterize psychological inflexibility: cognitive fusion, experiential avoidance, and the conceptualized self, in which people define their identity through their thoughts (*Hayes, 2015*).

In psychopathology, psychological inflexibility is considered a transdiagnostic etiological factor in the development and maintenance of psychological disorders and emotional difficulties (*Uğur, Kaya & Tanhan, 2021*). A person who has difficulty coping with unpleasant situations, through one or more of the six processes, is likely to be classified as psychologically inflexible, which in turn may contribute to the development of emotional disorders (*Orouji, Abdi & Chalabianloo, 2022*; *Tanhan, 2019*).

In the sporting arena, athletes who exhibit poor psychological flexibility may display fewer effective behaviors and are less likely to achieve optimal performance (*Moore, 2009*). Furthermore, psychological rigidity is associated with a greater number of distress

symptoms, including anxiety and depression (*Ruiz, 2010*), a trend also observed in various athletes (*Chen, Wu & Chang, 2017*; *Zhang et al., 2014*). Therefore, it is plausible to suggest that psychological inflexibility is linked to increased distress and impaired performance. However, it has also been observed that psychological flexibility positively influences the psychological skills necessary to attain optimal performance (*Fenoy-Castilla & Campoy-Ramos, 2012*).

When analyzing the relationship between perfectionism and psychological flexibility, it has been found that perfectionism is associated with cognitive processes such as performance monitoring and emotion regulation (*Ong et al., 2020*). In this context, individuals with a high degree of maladaptive perfectionism tend to respond rigidly to perfectionistic thoughts, avoiding uncomfortable emotions and engaging in inaction that is disconnected from their values (*Nguyen & Morris, 2024*). This experiential avoidance may serve as a significant factor contributing to the emergence of worry in the context of maladaptive perfectionism (*Santanello & Gardner, 2007*).

Consider perfectionism and psychological flexibility from this viewpoint is of particular clinical relevance since (operant) behaviors are considered to be under contextual control, suggesting they can be directly modified (*Gentili et al., 2019*). This will enable interventions to target these relevant factors, thereby facilitating the implementation of strategies by referees when confronted with stressful situations in their sporting activities. Given the lack of literature within the refereeing field addressing the relationships between these two coping factors, the present study focuses on football referees, whose roles are characterized by a high level of interaction and the need to process numerous stimuli (*MacMahon & Plessner, 2013*). The aim is to analyze the relationship between psychological inflexibility, perfectionism, and psychopathological symptomatology in soccer referees. We expect to find greater psychopathological symptomatology in amateur referees compared to semi-professional and professional referees. Additionally, higher levels of psychological inflexibility are predicted to be positively associated with elevated scores on psychopathological symptoms. Similarly, psychological inflexibility is expected to correlate positively with maladaptive perfectionism and negatively with adaptive perfectionism. Finally, we expect to see a negative relationship between adaptive perfectionism and psychopathological symptomatology and a positive relationship between maladaptive perfectionism and psychopathological symptomatology.

## MATERIALS & METHODS

### Participants

The eligibility criteria for this study were: (1) to be a soccer main referee, (2) over 18 years of age, (3) to have been a member of the Referees Committee of the Royal Spanish Soccer Federation for at least three years, (4) to be an active referee, and (5) to provide written informed consent.

The sample comprised 156 main referee (151 men, accounting for 96.8% of the sample) who were active members of the Referees Committee of the Royal Spanish Soccer Federation. Their mean age was 28.54 years (SD = 7.63). Regarding educational

attainment, 65.4% reported having university degrees, 26.9% had completed secondary education, and 7.7% had basic education. On average, participants had been federation members for 9.15 years (SD = 5.48). Regarding the categories of officials, 63.5% officiated in amateur leagues, while 36.5% were involved in semi-professional or professional leagues.

## Instruments

Information about sociodemographic variables (year of birth, sex, level of education), and arbitration variables (years of federation membership, arbitration category) was collected.

Psychopathological symptoms were assessed using the 45-item self-report instrument Symptom Assessment-45 Questionnaire (SA-45) (*Davison et al., 1997*), in the Spanish adaptation by *Sandín et al. (2008)*, which is a derived from the Symptom Checklist -SCL90- (*Derogatis & Cleary, 1977*). The questionnaire assesses the same dimensions as the SCL-90-R: hostility, somatization, depression, obsession-compulsion, anxiety, interpersonal sensitivity, phobic anxiety, paranoid ideation and psychoticism. Participants are asked to answer each item (*e.g.*, the belief that another person can control one's thoughts) by indicating the frequency with which they have experienced each of the 45 symptoms during the past week, between 0 ('not at all') and 4 ('very or extremely'). The test provides a total score as well as individual scores for each referred subscale. Evidence in support of its reliability and validity has been reported for both the English version (*Davison et al., 1997*) and the Spanish version (*Sandín et al., 2008*). The reliability demonstrated in this study is a Cronbach's $\alpha = .965$.

The Acceptance and Action Questionnaire (AAQ-II) (*Bond et al., 2011*), adapted to Spanish by *Ruiz et al. (2013)*, was used to assess psychological inflexibility/flexibility. This is a 7-item questionnaire concerned with how the individual relates to their internal events (*e.g.*, thoughts, feelings, emotions, and memories) and to what extent they perceive these events as barriers to leading the life they wish. Participants respond using a Likert-type scale ranging from 1 ('never true') to 7 ('always true') to indicate the extent of their belief in the statements (*e.g.*, Worries get in the way of my success). Low scores on the questionnaire indicate greater psychological flexibility, while high scores indicate greater inflexibility. The test used in this study has shown high internal consistency (Cronbach's $\alpha = .939$). To examine the relationship between the level of psychological flexibility and the other variables, participants were categorized using a Cluster analysis, resulting in three groups (low psychological inflexibility, medium psychological inflexibility and high psychological inflexibility).

The Frost Multidimensional Perfectionism Scale (MPS) (*Frost et al., 1990*), was used to assess perfectionism; in its Spanish adaptation by *Carrasco, Belloch & Perpiñá (2010)*. The MPS is a 35-item self-report instrument where participants respond on a Likert-type scale ranging from 1 ('strongly disagree') to 5 ('strongly agree') to a set of statements (*e.g.*, If I fail partly, it is as bad as being a complete failure). The Spanish version has enable the identification of four factors: MPS-F1—fear of making mistakes (concern over mistakes and doubts about actions); MPS-F2—external influences (parental expectations and parental criticism); MPS-F3—expectations of achievement (personal standards and two items of concern over mistakes); MPS-F4—organization. These factors can be grouped into

an MPS-MALA—maladaptive perfectionism (Factor 1 and Factor 2) and MPS-ADAP—adaptive perfectionism (Factor 3 and Factor 4). The internal consistency of the two types of perfectionism in this study was: MPS-MALA—maladaptive perfectionism ($\alpha = .926$) and MPS-ADAP—adaptive perfectionism ($\alpha = .838$).

## Institutional Review Board Statement

All procedures were in accordance with the ethical standards of the responsible committee on human experimentation (institutional and national) and the Declaration of Helsinki of 1975, revised in 2013. Approved by the Andalusian Ethics Committee of Biomedical Research (Evaluation Committee of Huelva. Act: 05/24. Date of approval:14-May-2024, Internal Code: SICEIA-2024-001020, Study code: 21071). Informed Consent Statement: Informed consent was obtained from all individual participants included in the study.

## Procedure

Data collection was conducted online from 15th May 2024 to 31st August 2024. The Referees Committee of the Royal Spanish Soccer Federation was contacted, and all Territorial Committees were sent information on the study, requesting their collaboration by disseminating the address needed to access the online questionnaires among the active referees in the federation. The participants had to accept informed consent in order to complete the tests; in the online test, access was not allowed if the option to accept consent was not chosen.

This cross-sectional, anonymous, online study was conducted in accordance with the ethical standards of the responsible committee on human experimentation (institutional and national) and the Declaration of Helsinki of 1975, revised in 2013.

## Data analysis

An a priori power analysis was conducted using G*Power 3 (*Faul et al., 2007*) to determine the minimum sample size required for testing the study hypothesis. The results indicated that a sample size of $n = 147$ was necessary for Student's $t$-test for independent groups in order to achieve 95% power to detect a medium effect at a significance level of $\alpha = .05$.

Descriptive analyses (frequencies, percentages, means, and standard deviation) were conducted to characterize the main research variables. The normality of the variables is confirmed using the Kolmogorov–Smirnov test. The preconditions required for each of the tests considered in this research were assessed (*e.g.*, homoscedasticity, collinearity -variance inflation factor, VIF-, *etc.*). The Mann–Whitney U test, for comparisons between two groups, and the Kruskal–Wallis test, for comparisons involving more than two groups, were used to analyse variables that did not conform to normality; being the most robust option in each case. The effect size estimate in the Mann–Whitney U-test was calculated using the formulation r $= Z/\sqrt{n}$, (<0.099—insignificant effect size; 0.100–0.299—small effect size; 0.300–0.499—medium effect size; >0.500—large effect size). The reliability of the tests was calculated using Cronbach's alpha ($\alpha$). To calculate the effect size in the Student's t test, Cohen's d has been considered ($d < 0.2$—small effect size; $d = 0.2$ to 0.8—medium effect size and $d > 0.8$—large effect size). Bonferroni tests were used for *post hoc* ANOVA comparisons. The effect size was calculated using Eta Squared $\eta^2$ ($0.01 \leq \eta2$

$<0.06 =$ a small effect size, $0.06 \leq \eta^2 <0.14 =$ a medium effect size, and $\eta^2 \geq 0.14 =$ a large effect size). Associations between the variables were analyzed by Pearson and Spearman's Rho correlations and hierarchical linear regression analysis was employed to determine the predictors of psychopathological symptoms. A cluster analysis was conducted to group participants based on their levels of psychological inflexibility, allowing for subsequent comparisons across the dependent variables. It is important to note that there were no missing data, as all items were required to be completed in order to proceed with the instrument. Analyses were conducted using the SPSS statistical package (IBM version 25.0; SPSS Inc Armonk, NY, USA).

## RESULTS

According to the analyses conducted, using G*Power-3, the obtained sample size of $n = 156$ was deemed adequate for testing the study hypotheses. The sample consisted of 156 main referee, with a mean age of 28.48 years (SD = 7.72) for men and 30.20 years (SD = 4.08) for women. A significant age difference was found between amateur officials ($M = 27.36$ years, SD = 8.146) and semi-professional/professional officials ($M = 30.58$ years, SD = 6.193), $t_{(154)} = 2.580$, $p = .011$, with a medium effect size ($d = 0.45$). Additionally, significant differences were found in the duration of federation membership between amateur referees ($M = 7.77$ years, SD = 4.825) and semi-professional/professional referees ($M = 11.54$ years, SD = 5.766), $t_{(154)} = 4.378$, $p < .001$, also with a medium effect size ($d = 0.71$).

A Kolmogorov–Smirnov test was conducted to assess the normality of the distribution of the variables. The results indicated a normal distribution for psychological inflexibility ($Z = 0.823$, $p = .058$), MPS-MALA (maladaptive perfectionism) ($Z = 0.920$, $p = .365$), MPS-F1 (fear of making mistakes) ($Z = 0.783$, $p = .572$), MPS-F2 (external influences) ($Z = 1.152$, $p = .141$), MPS-ADAP (adaptive perfectionism) ($Z = 0.725$, $p = .669$), MPS-F3 (expectations of achievement) ($Z = 0.866$, $p = .441$), and MPS-F4 (organization) ($Z = 1.092$, $p = .184$). However, the psychopathological symptoms and their subscales did not conform to a normal distribution (SA-45: $Z = 1.508$, $p = .021$; depression: $Z = 1.894$, $p = .002$; hostility: $Z = 2.548$, $p < .001$; interpersonal sensitivity: $Z = 2.117$, $p < .001$; somatization: $Z = 1.424$, $p = .035$; anxiety: $Z = 1.928$, $p = .001$; psychoticism: $Z = 2.552$, $p < .001$; obsession-compulsion: $Z = 1.848$, $p = .002$; phobic anxiety: $Z = 3.706$, $p < .001$; paranoid ideation: $Z = 1.595$, $p = .012$).

As evident in Table 1, no statistically significant differences were observed in psychological inflexibility and overall psychopathological symptomatology between amateur and semi-professional/professional officials, except for the paranoid ideation subscale, where amateurs score higher ($d = 0.4$), with a medium effect size. Significant differences were observed in the total score of maladaptive perfectionism ($d = 0.4$) and specifically in MPS-F2 (external influences), where amateur referees scored higher ($d = 0.5$), both with medium effect sizes. Regarding adaptive perfectionism, differences were found in the total score ($d = 0.4$) and MPS-F3 (expectations of achievement), where amateur referees scored higher ($d = 0.4$), both with medium effect sizes.

**Table 1 Symptom Assessment-45 questionnaire scores, multidimensional perfectionism scale (MPS) and AAQ-II-Psychological inflexibility according to referee category.** The mean scores on the questionnaires, according to the professional level of the referee, are indicated.

| | Total 156 | Amateur 99 (63.5) | Semiprofessional/ Professional 57 (36.5) | Z U Mann–Whitney | p |
|---|---|---|---|---|---|
| SA-45-TOTAL | 33.71 (26.07) | 35.13 (25.85) | 31.25 (26.48) | −1.222 | 0.222 |
| Depression | 4.46 (4.25) | 4.78 (4.59) | 3.89 (3.54) | −0.861 | 0.389 |
| Hostility | 2.70 (3.12) | 2.81 (3.24) | 2.51 (2.93) | −0.528 | 0.597 |
| Interpersonal sensitivity | 4.06 (3.98) | 4.33 (4.03) | 3.60 (3.88) | −1.349 | 0.177 |
| Somatisation | 4.33 (3.58) | 4.35 (3.34) | 4.30 (3.99) | −0.670 | 0.503 |
| Anxiety | 3.94 (3.62) | 3.96 (3.62) | 3.91 (3.66) | −0.219 | 0.827 |
| Psychoticism | 2.50 (2.85) | 2.67 (2.95) | 2.21 (2.66) | −1.159 | 0.246 |
| Obsession-Compulsion | 5.47 (3.83) | 5.72 (3.79) | 5.04 (3.89) | −1.167 | 0.243 |
| Phobic anxiety | 1.47 (2.61) | 1.27 (2.28) | 1.81 (3.09) | −1.143 | 0.253 |
| Paranoid ideation | 4.78 (3.83) | 5.24 (3.88) | 3.98 (3.64) | −2.185 | 0.029 |
| | | | | $t_{(df=154)}$ | |
| MPS-MALA | 47.44 (9.99) | 48.80 (9.64) | 45.09 (10.23) | 2.263 | 0.025 |
| MPS-F1 | 24.83 (8.53) | 25.75 (8.71) | 23.25 (8.03) | 1.777 | 0.078 |
| MPS-F2 | 18.83 (7.29) | 20.04 (7.72) | 16.72 (5.97) | 2.800 | 0.006 |
| MPS-ADAP | 43.66 (14.19) | 45.79 (14.61) | 39.96 (12.74) | 2.509 | 0.013 |
| MPS-F3 | 26.06 (7.10) | 26.98 (6.76) | 24.47 (7.45) | 2.148 | 0.033 |
| MPS-F4 | 21.38 (4.77) | 21.82 (4.83) | 20.61 (4.62) | 1.524 | 0.129 |
| AAQ-II | 18.57 (9.00) | 18.9 (9.37) | 17.98 (8.37) | 0.618 | 0.538 |

**Notes.**
For quantitative variables M(SD) and categorical variables n(%).
Z, Statistic of the Mann–Whitney U-test; SA-45, Symptom Assessment-45 Questionnaire; MPS-MALA, Maladaptive Perfectionism; MPS-F1, Fear of Making Mistakes; MPS-F2, External Influences; MPS-ADAP, Adaptive Perfectionism; MPS-F3, Expectations of Achievement; MPS-F4, Organization; AAQ-II, the Acceptance and Action Questionnaire (Psychological Inflexibility).

Grouping participants into three clusters based on their psychological inflexibility scores revealed significant differences in the total score of psychopathological symptomatology and across its subscales (see Table 2). These differences were detected using the Kruskal–Wallis test and were highly statistically significant.

When conducting *post hoc* comparisons using the Bonferroni test on the scales assessing perfectionism, significant differences were found among the three groups in maladaptive perfectionism (MPS-PF-MALA), with a large effect size ($\eta^2 = 0.17$): a < c ($p < 0.001$), a < b ($p = 0.032$), and c < b ($p = 0.026$). Similarly, significant differences were found for the MPS-F1 subescale (fear of making mistakes), also with a large effect size ($\eta^2 = 0.19$): a < c ($p < 0.001$), a < b ($p = 0.008$), and c < b ($p = 0.002$). However, for the MPS-F2 subscale (external influences), no differences were found between the three groups, a = c ($p = 0.070$), a = b ($p = 0.392$), and c = b ($p = 0.674$). In contrast, no significant group differences were observed among the three groups in adaptive perfectionism (MPS-ADAP): a = c ($p = 0.345$), a = b ($p = 0.909$), and c = b ($p = 1.00$). Neither in the expectations of achievement subscale (MPS-F3), observing that a = c ($p = 0.172$), a = b ($p = 0.476$), and c = b ($p = 1.00$). Likewise, no differences were found in the organization factor (MPS-F4): a = c ($p = 1.00$), a = b ($p = 1.00$), and c = b ($p = 1.00$).

**Table 2 Symptom Assessment-45 Questionnaire scores, Multidimensional Perfectionism Scale (MPS) according to AAQ-II-Psychological inflexibility cluster.** The mean scores on the questionnaires according to the clusters in Psychological Inflexibility are indicated.

| | Low psychological inflexibility (a) 83(53.2) | Medium psychological inflexibility (b) 55(35.3) | High psychological inflexibility (c) 18(11.5) | $F_{(2,155)}$ | $p$ |
|---|---|---|---|---|---|
| MPS-MALA | 39.73 (12.94) | 45.75 (13.36) | 55.39 (15.09) | 11.214 | <0.001 |
| MPS-F1 | 22.07 (7.131) | 26.18 (8.28) | 33.44 (8.83) | 17.195 | <0.001 |
| MPS-F2 | 17.66 (6.92) | 19.56 (6.91) | 21.94 (9.12) | 3.066 | 0.049 |
| MPS-ADAP | 46.34 (9.79) | 48.13 (10.92) | 50.44 (7.17) | 1.458 | 0.236 |
| MPS-F3 | 25.05 (6.85) | 26.78 (7.48) | 28.56 (6.50) | 2.277 | 0.106 |
| MPS-F4 | 21.29 (4.62) | 5.12 (0.69) | 21.89 (4.60) | 0.117 | 0.889 |
| | | | | Kruskal–Wallis $\chi^2_{(df=2)}$ | |
| SA-45-TOTAL | 19.57 (14.05) | 42.25 (20.74) | 72.83 (32.29) | 59.357 | <0.001 |
| Depression | 2.12 (20.9) | 6.13 (3.78) | 10.11 (5.34) | 62.481 | <0.001 |
| Hostility | 1.61 (2.01) | 3.24 (3.10) | 6.06 (4.48) | 24.578 | <0.001 |
| Interpersonal sensitivity | 2.07 (20.7) | 5.09 (3.49) | 10.11 (4.86) | 49.673 | <0.001 |
| Somatisation | 3.06 (2.63) | 5.36 (3.48) | 7.06 (5.06) | 22.439 | <0.001 |
| Anxiety | 1.96 (1.85) | 5.38 (3.31) | 8.67 (4.33) | 58.500 | <0.001 |
| Psychoticism | 1.37 (1.67) | 2.89 (2.46) | 6.50 (4.22) | 38.523 | <0.001 |
| Obsession-Compulsion | 3.66 (2.69) | 6.67 (3.51) | 10.11 (4.01) | 44.550 | <0.001 |
| Phobic anxiety | 0.73 (1.43) | 1.45 (2.19) | 4.89 (4.66) | 25.277 | <0.001 |
| Paranoid ideation | 2.96 (2.49) | 6.04 (3.64) | 9.33 (4.34) | 42.849 | <0.001 |

**Notes.**

For quantitative variables M(SD) and categorical variables n(%).

MPS-MALA, Maladaptive Perfectionism; MPS-F1, Fear of Making Mistakes; MPS-F2, External Influences; MPS-ADAP, Adaptive Perfectionism; MPS-F3, Expectations of Achievement; MPS-F4, Organization; SA-45, Symptom Assessment-45 Questionnaire.

Table 3 presents correlations between the different variables. Correlations between subscales of the same test (SA-45 and MPS) were omitted as they were not relevant to the goals of this study; however, all correlations within each test were highly significant.

Psychological inflexibility showed significant correlations with all tests and subscales except the organization factor. Similarly, the organization factor did not correlate significantly with the total psychopathological symptomatology score or any of its subscales. However, the expectations of achievement factor showed significant correlations with all mental health subscales except for phobic anxiety. Specifically, the total adaptive perfectionism score was significantly correlated with the total SA-45 test score and its subscales, except for obsession-compulsion and phobic anxiety.

In the case of maladaptive perfectionism, both the total score and the fear of making mistakes subscale showed significant correlations with the SA-45 total score and the nine subscales. However, the external influences factor did not correlate significantly with the depression, somatization, or phobic anxiety subscales.

Linear regression models (see Table 4) were generated with psychopathological symptomatology scores as the predicted variable and psychological inflexibility, adaptive perfectionism, and maladaptive perfectionism as predictor variables. After checking the basic requirements of regression models, such as homoscedasticity and collinearity using

**Table 3  Bivariate correlation between scores on psychological inflexibility, perfectionism, and psychopathological symptoms.** The correlations between the different variables that have been evaluated are indicated.

|  |  | 1 | 2 | 3 | 4 | 5 | 6 | 7 |
|---|---|---|---|---|---|---|---|---|
| 1 | AAQ-II | 1 | | | | | | |
| 2 | MPS-MALA | 0.432/<0.001 | | | | | | |
| 3 | MPS-F1 | 0.503/<0.001 | | | | | | |
| 4 | MPS-F2 | 0.253/0.001 | | | | | | |
| 5 | MPS-ADAP | 0.177/0.027 | | | | | | |
| 6 | MPS-F3 | 0.241/0.002 | | | | | | |
| 7 | MPS-F4 | 0.013/0.868 | | | | | | |
| 8 | SA-45-TOTAL | 0.725/<0.001 | 0.403/<0.001 | 0.505/<0.001 | 0.210/0.008 | 0.274/0.001 | 0.345/<0.001 | 0.056/0.484 |
| 9 | Depression | 0.727/<0.001 | 0.295/<0.001 | 0.390/<0.001 | 0.149/0.064 | 0.176/0.028 | 0.248/0.002 | −0.032/0.695 |
| 10 | Hostility | 0.450/<0.001 | 0.336/<0.001 | 0.394/<0.001 | 0.194/0.015 | 0.258/0.001 | 0.333/<0.001 | 0.023/0.772 |
| 11 | Inter. Sensitivity | 0.652/<0.001 | 0.344/<0.001 | 0.436/<0.001 | 0.187/0.019 | 0.224/0.005 | 0.298/<0.001 | 0.020/0.800 |
| 12 | Somatisation | 0.450/<0.001 | 0.186/0.020 | 0.278/<0.001 | 0.065/0.420 | 0.176/0.028 | 0.184/0.022 | 0.117/0.147 |
| 13 | Anxiety | 0.727/<0.001 | 0.356/<0.001 | 0.443/<0.001 | 0.181/0.024 | 0.264/0.001 | 0.320/<0.001 | 0.087/0.282 |
| 14 | Psychoticism | 0.600/<0.001 | 0.447/<0.001 | 0.496/<0.001 | 0.291/<0.001 | 0.245/0.002 | 0.319/<0.001 | 0.052/0.518 |
| 15 | Obses-Compuls | 0.622/<0.001 | 0.323/<0.001 | 0.397/<0.001 | 0.169/0.035 | 0.145/0.072 | 0.214/0.007 | −0.028/0.732 |
| 16 | Phobic anxiety | 0.398/<0.001 | 0.225/0.005 | 0.276/<0.001 | 0.113/0.161 | 0.081/0.315 | 0.148/0.065 | −0.044/0.587 |
| 17 | Paran ideation | 0.604/<0.001 | 0.388/<0.001 | 0.471/<0.001 | 0.215/0.007 | 0.309/<0.001 | 0.366/<0.001 | 0.093/0.247 |

**Notes.**

r/p, Pearson Correlation/Significance (Spearman's Rho in the SA-45) (all correlations of subscales within each instrument (SA-45 and MPS) have been removed as they are irrelevant to the objective, although they are all highly significant); MPS-MALA, Maladaptive Perfectionism; MPS-F1, Fear of Making Mistakes; MPS-F2, External Influences; MPS-ADAP, Adaptive Perfectionism; MPS-F3, Expectations of Achievement; MPS-F4, Organization; AAQ-II, the Acceptance and Action Questionnaire (Psychological Inflexibility); SA-45, Symptom Assessment-45 Questionnaire.

the variance inflation factor (VIF), three significant models were identified. Maintaining, in model 1, an adequate VIF in the case of psychological inflexibility (VIF = 1); in model 2, psychological inflexibility showed a VIF = 1.229 and maladaptive perfectionism a VIF = 1.229; being in model 3 where psychological inflexibility was found a VIF = 1.242 and a greater maladjustment for maladaptive perfectionism (VIF = 1.826) and adaptive perfectionism (VIF = 1.534).

In the first model, psychological inflexibility explained 51.2% of the variance in psychopathological symptomatology, with a predictive power ($\beta$) of 0.716 and a semi-partial correlation of 0.716. In the second model, including maladaptive perfectionism significantly increased the explanatory capacity to 53.7%. However, the predictive power of psychological inflexibility decreased to $\beta = 0.640$, while maladaptive perfectionism yielded a $\beta = 0.176$. Semi-partial correlations in the second model were 0.577 for psychological inflexibility and 0.159 for maladaptive perfectionism.

In the third model, the inclusion of adaptive perfectionism significantly increased the overall explanatory power to 54.0%. However, the predictive capacity of adaptive perfectionism fell short of significance ($\beta = 0.069$), while maladaptive perfectionism also lost its significance as a predictor in the model. In contrast, psychological inflexibility showed slightly higher predictive power compared to the second model ($\beta = 0.646$). Examination of semi-partial correlations revealed values of 0.579 for psychological inflexibility, 0.099 for maladaptive perfectionism, and 0.055 for adaptive perfectionism.

**Table 4 Lineal regression analysis, taking psychopathological symptoms (SA-45) as the predicted variable and psychological inflexibility and adaptive-maladaptive perfectionism as predictor variables, in soccer referees.** The regression-prediction analyses of the variable psychopathological symptomatology are shown; according to the variables perfectionism and psychological inflexibility.

| | B | 95% IC | | β | t | p | $R^2$ | $\Delta R^2$ | p | F | p |
|---|---|---|---|---|---|---|---|---|---|---|---|
| **Model 1** | | | | | | | 0.512 | 0.512 | <0.001 | $F_{(1,155)} = 161.537$ | <0.001 |
| AAQ-II | 2.071 | 1.749 | 2.393 | 0.716 | 12.710 | <0.001 | | | | | |
| **Model 2** | | | | | | | 0.537 | 0.025 | 0.004 | $F_{(2,155)} = 88.776$ | <0.001 |
| AAQ-II | 1.851 | 1.503 | 2.200 | 0.640 | 10.488 | <0.001 | | | | | |
| MPS-PF-MALA | 0.323 | 0.102 | 0.544 | 0.176 | 2.886 | 0.004 | | | | | |
| **Model 3** | | | | | | | 0.540 | 0.003 | 0.315 | $F_{(3,155)} = 59.528$ | <0.001 |
| AAQ-II | 1.870 | 1.519 | 2.220 | 0.646 | 10.536 | <0.001 | | | | | |
| MPS-PF-MALA | 0.244 | −0.025 | 0.514 | 0.133 | 1.792 | 0.075 | | | | | |
| MPS-PF-ADAP | 0.179 | −0.172 | 0.530 | 0.069 | 1.008 | 0.315 | | | | | |

**Notes.**
AAQ-II, The Acceptance and Action Questionnaire (Psychological Inflexibility); MPS-PF-MALA, Maladaptive Perfectionism; MPS-PF-ADAP, Adaptive Perfectionism.

## DISCUSSION

This study examined the relationship between psychological inflexibility, perfectionism, and psychopathological symptomatology in soccer referees. The main results indicated that there were no significant differences between amateur and semi-professional/professional referees in psychological inflexibility and overall psychopathological symptoms, except in the case of paranoid ideation, where amateurs scored higher. Regarding perfectionism, differences emerged in maladaptive perfectionism, particularly in the dimension of external influences, with amateurs again scoring higher. In terms of adaptive perfectionism, significant differences were found in total scores and achievement expectations, with higher scores among amateurs. Psychological inflexibility and maladaptive perfectionism demonstrated strong predictive power for psychopathological symptomatology, whereas adaptive perfectionism did not significantly predict symptomatology.

Our first hypothesis predicted that amateur referees would show greater psychopathological symptomatology compared to semi-professional and professional referees. However, our findings did not fully support this hypothesis, as no significant differences were observed between these two groups, with the exception of paranoid ideation. This lack of differences contradicts the existing literature demonstrating that officials in professional categories typically obtain lower scores on mental health indicators compared to their amateur counterparts (*Carson et al., 2020*; *Lima et al., 2023*). One possible explanation for the absence of differences in psychopathological symptoms between amateurs and semi-professionals/professionals is the low representation of professional referees in the sample. In this regard, the pressure perceived by the referee may be very similar in the amateur and semi-professional contexts, compared to that perceived in professional leagues; where stakes, infrastructures (distance from the public, institutional relations, security, *etc.*) and resources are much higher (*Gouttebarge et al., 2017*).

As mentioned, significant differences were found in paranoid ideation, with amateur referees scoring higher. This can be attributed to increased preoccupation with aggression,

limited resources and security at the lower levels of refereeing (*Cuskelly & Hoye, 2013*). These contextual factors may be particularly relevant in light of the components of the paranoid ideation subscale, which include attributing problems to others, distrusting people, feeling scrutinized or talked about, and perceiving one's achievements as unacknowledged (*Davison et al., 1997*; *Sandín et al., 2008*). In this sense, for paranoid ideation, age may have been a factor of vulnerability as has been indicated in the literature (*Carson et al., 2020*; *Fonseca-Pedrero et al., 2009*; *Lima et al., 2022*; *Scott et al., 2009*).

As a second hypothesis, it was predicted that greater psychological inflexibility would be positively associated with higher scores on psychopathological symptoms. Our findings fully support this hypothesis and align with previous research. In the general population, psychological rigidity has been strongly linked to distress, anxiety, depression, and other mental health issues (*Arbinaga & Cantón, 2013*; *Ruiz et al., 2013*), with the associated behavioral patterns often hindering mental health improvement and potentially exacerbating problems (*Trompetter et al., 2015*; *Wicksell, Olsson & Hayes, 2010*). In the context of clinical psychology, psychological inflexibility is considered a transdiagnostic etiological factor in the development and maintenance of psychological disorders and emotional difficulties (*Hayes, 2015*; *Uğur, Kaya & Tanhan, 2021*). When individuals experience difficulties in coping with situations through the processes that characterize psychological inflexibility, they are likely to adopt generalized and rigid behavioral patterns that lack value-guided action; this may lead to the onset of emotional disorders (*Tanhan, 2019*). In contrast, psychological flexibility facilitates the resolution of problems through adaptive responses (*Hayes, 2015*). Therefore, psychological flexibility is associated with regulation and adaptive coping processes that reflect better psychological health (*Kashdan & Rottenberg, 2010*).

Therefore, intervening on the construct of psychological (in)flexibility is necessary, as there is considerable evidence linking it to a broad spectrum of psychological disorders, particularly those characterized by an avoidant reaction style (*Bond et al., 2011*; *Levin et al., 2014*). Numerous studies support the idea that psychological inflexibility plays a mediating role between stress and depression, somatization and anxiety (*Arslan et al., 2021*), as well as in the relationship between fear of negative evaluation and psychological vulnerability (*Uğur, Kaya & Tanhan, 2021*). *Levin et al. (2014)* examined psychological inflexibility as a transdiagnostic process across various psychological disorders.

Similarly, in sports, low psychological flexibility has been linked to reduced behavioral effectiveness and missed opportunities for optimal performance (*Moore, 2009*). Additionally, significant associations have been observed between psychological inflexibility and symptoms of psychological disorders among athletes (*Chen, Wu & Chang, 2017*; *Zhang et al., 2014*).

On the other hand, our third hypothesis anticipated that psychological inflexibility would show a positive association with maladaptive perfectionism and a negative association with adaptive perfectionism. Our data partially support this hypothesis, as positive correlations were observed in both cases, indicating that higher scores in psychological inflexibility correspond to higher scores in both adaptive and maladaptive perfectionism. In this

regard, psychological inflexibility has been found to be related to perfectionism (*Habibi-Asgarabad et al., 2023*; *Miles, Nedeljkovic & Phillipou, 2023*). The literature has shown that a high degree of maladaptive perfectionism facilitates psychologically inflexible behaviors, characterized by rigid responses to thoughts, feelings, and bodily sensations, leading to psychological distress and avoidance behaviors (*Crosby et al., 2013*). A central feature of psychological inflexibility is avoidance, and perfectionists are often inclined to employ unhelpful avoidance strategies, such as experiential avoidance (*Santanello & Gardner, 2007*), avoidant coping (*Noble, Ashby & Gnilka, 2014*), and emotional suppression (*Richardson, Rice & Devine, 2014*), particularly in response to challenges. Individuals with higher self-critical or maladaptive perfectionism are more likely to experience need dissatisfaction, which is explained by the fact that these individuals experience more depressive symptoms (*Levine, Andrade & Koestner, 2022*). The results suggest that people with higher self-critical perfectionism are less flexible when things do not go according to plan.

Our fourth hypothesis predicted that psychopathological symptomatology would show a negative relationship with adaptive perfectionism and a positive relationship with maladaptive perfectionism. Our findings partially support this hypothesis, as the expected negative relationship between psychopathological symptomatology scores and adaptive perfectionism was not found, while a positive relationship was observed with maladaptive perfectionism. These relationships have been previously reported in the work of *Stoeber & Otto (2006)*, where perfectionism was linked to various psychopathologies. Those with higher adaptive perfectionism, related to personal standards, experience greater overall need satisfaction, which in turn is associated with a reduction in depressive symptoms (*Levine, Andrade & Koestner, 2022*). In the context of sports, while perfectionism is often seen as a motivator, it can also present challenges or difficulties (*Hill et al., 2015*). Perfectionism is associated with several cognitive processes including performance monitoring and emotion regulation (*Ong et al., 2020*). Specifically, perfectionism has been associated with anxiety (*Schaal et al., 2011*; *Stoeber et al., 2007*), depression (*Crocker et al., 2014*; *Gorczynski, Coyle & Gibson, 2017*; *Gulliver et al., 2015*; *Nixdorf, Frank & Beckmann, 2016*; *Schaal et al., 2011*; *Tashman, Tenenbaum & Eklund, 2010*; *Wolanin et al., 2016*) and stress (*Crocker et al., 2014*; *Schaal et al., 2011*; *Tashman, Tenenbaum & Eklund, 2010*).

Nevertheless, the results of this paper do reinforce (*Hill et al., 2008*) in stating that problems were mainly associated with maladaptive perfectionism, but adaptive perfectionism could be observed to generate distress. This distinction can be understood by considering that adaptive perfectionism emphasizes behavioral organization and goal setting to enhance sporting performance, whereas maladaptive perfectionism focuses on responses to errors or failure to achieve goals. Thus, perfectionism should be viewed as a vulnerability factor characterized by cognitive rigidity and behavioral inflexibility (*Delor, Douilliez & Philippot, 2019*).

The results found in the literature linking psychological inflexibility, psychopathological symptoms, and perfectionism have been confirmed within a population group, such as referees and the sports community, for which no previous data existed. Based on these findings, referees could benefit from interventions that develop strategies and skills to

mitigate the impact of maladaptive strategies (*e.g.*, cognitive fusion, experiential avoidance). Acceptance and Commitment Therapy (ACT), with its emphasis on psychological flexibility, has proven effective in addressing perfectionism (*Taghavizade-Ardakani et al., 2019*). Studies analyzing neurological data from a randomized controlled trial on clinical perfectionism (*Ong et al., 2020*) support the change processes outlined by the theoretical framework of ACT. These findings provide initial support for the feasibility and efficacy of a process-based approach.

Among the limitations of this study are the sample size and gender balance, since female participation was low. The low participation of women limits the generalizability of the results to the population of female referees. Another factor that may restrict the generalizability is the lack of consideration of assistant referees. Additionally, the design and methodology used preclude establishing causal relationships, while reliance on self-reported data may introduce response biases. Future research would benefit from greater control over refereeing contexts. Designs that evaluate officials at different points in the season, considering factors such as injuries, travel demands, and number and significance of matches officiated, would help to enhance our understanding of the factors involved in the psychological disorders associated with this profession. Finally, when studying mental health in this population, it is also important consider non-sporting activities, as well as individual's medical, psychiatric and clinical-psychological history.

## ACKNOWLEDGEMENTS

To the Technical Committees of Territorial Referees who have collaborated and to all the referees who have completed the evaluation process.

### Funding
This work has received funding from EPIT-UHU for data collection and the PAIDI financial support to the Research Group "Psychology and Emerging Social Problems" SEJ-451 for translation. There was no additional external funding received for this study. The funders had no role in study design, data collection and analysis, decision to publish, or preparation of the manuscript.

### Grant Disclosures
The following grant information was disclosed by the authors:
EPIT-UHU for data collection and the PAIDI financial support to the Research Group "Psychology and Emerging Social Problems" SEJ-451 for translation.

### Competing Interests
The authors declare there are no competing interests.

## Author Contributions

- Félix Arbinaga conceived and designed the experiments, performed the experiments, analyzed the data, prepared figures and/or tables, authored or reviewed drafts of the article, and approved the final draft.
- Emilio Moreno San Pedro conceived and designed the experiments, performed the experiments, analyzed the data, prepared figures and/or tables, authored or reviewed drafts of the article, and approved the final draft.
- María-Isabel Mendoza-Sierra conceived and designed the experiments, performed the experiments, analyzed the data, prepared figures and/or tables, authored or reviewed drafts of the article, and approved the final draft.

## Human Ethics

The following information was supplied relating to ethical approvals (*i.e.*, approving body and any reference numbers):

This research was approved by the Andalusian Ethics Committee of Biomedical Research (Evaluation Committee of Huelva-SICEIA) (Internal Code: SICEIA-2024-001020, Study code: 21071).

## Data Availability

The raw data is available in the Supplemental File.

All anonymized raw participant data are available at the Open Science Framework: Ibarzabal, Félix A. 2025. ''Referee- Psychopathological-Psychol-Inflexibil-Perfect.'' OSF. May 11. doi: 10.17605/OSF.IO/MZJSC.

## Supplemental Information

Supplemental information for this article can be found online at http://dx.doi.org/10.7717/peerj.19790#supplemental-information.

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
