# Peer review of "Psychopathological symptoms in soccer referees: the role of psychological inflexibility and perfectionism"

_PeerJ, doi:10.7717/peerj.19790_

## Round 0.1 · original submission · Major Revisions

The reviewers have made several suggestions that will help to improve your manuscript significantly. Please address those in your response with submission of your next draft.

·

Basic reporting

*Although previous studies are mentioned, the introduction could benefit from a more in-depth discussion of how psychological inflexibility and perfectionism have been studied in other sports contexts (e.g., among athletes or coaches). This would help position the study more effectively within the existing literature.
*Including more details on contradictory findings or gaps in the literature would further justify the need for this study. For example, why is it important to study these constructs specifically in referees rather than just in athletes?
*The introduction could place greater emphasis on the unique characteristics of referees as a study group. For instance, what differentiates referees from athletes in terms of stress, pressure, or psychological resources? This would help further justify the focus on this group.
*Although concepts such as psychological inflexibility and perfectionism are mentioned, the introduction could benefit from a more detailed explanation of how these constructs are theoretically related to psychopathological symptoms. For example, what psychological mechanisms underlie these relationships?
*Some sentences are somewhat long or complex, which may hinder readability. For example:
"In a study by Kilic et al. (2018) conducted among soccer referees across eight European countries, 5.9% reported symptoms of distress, 11.8% reported anxiety/depression, 9.1% reported sleep disturbances, and 16.5% reported adverse alcohol use."
Could be revised as:
"Kilic et al. (2018) conducted a study among soccer referees from eight European countries. They found that 5.9% reported distress, 11.8% reported anxiety/depression, 9.1% reported sleep disturbances, and 16.5% reported adverse alcohol use."
*There are some inconsistencies in style, such as the use of commas before "and" in lists (e.g., "hostility, somatization, depression, obsession-compulsion, anxiety, interpersonal sensitivity, phobic anxiety, paranoid ideation, and psychoticism"). In American English, it is common to use a comma before "and" (known as the "Oxford comma"), while in British English it is not always used. It is recommended to choose a style and maintain consistency.
*Certain words and phrases are frequently repeated (e.g., "mental health problems," "referees"). Using synonyms or variations (e.g., "psychological distress," "officials") could make the text more dynamic and less repetitive.
*"Result" should be "Results" in the abstract.
*"So on" should be replaced with "Thus" or "Therefore" for greater formality.
*"The idea that another person can control their thoughts" could be clearer as "The idea that someone else can control their thoughts."
*Overall, the tone is appropriate for a scientific article, but in some sections, the language could be more formal. For example:
"So on, in the sports context, perfectionism has been associated with anxiety..."
Could be revised to:
"Therefore, in the sports context, perfectionism has been associated with anxiety..."

Experimental design

Although the use of GPower for sample size calculation is mentioned, the specific parameters used (effect size, statistical power, alpha level) are not provided. Including these details would enhance transparency.
*The sample of 156 participants is adequate; however, the low representation of women (only 5 out of 156) limits the generalizability of the findings to female referees. This should be explicitly mentioned as a limitation.
*The description of the SA-45 could be expanded to clarify whether it was administered in its entirety or if specific subscales were prioritized. This is relevant because the results focus on particular subscales (e.g., paranoid ideation).
*For the AAQ-II, it is mentioned that participants were categorized into tertiles based on their psychological inflexibility scores. However, there is no justification for why tertiles were chosen over other categorization methods (e.g., quartiles or clinical cut-off points). A brief explanation would strengthen this decision.
*It is stated that data collection took place between October 2023 and January 2024, but ethical approval is mentioned as being granted in May 2024. This chronological discrepancy should be clarified to ensure ethical compliance.
*It would be useful to include information on the response rate or how many referees were invited to participate. This would provide context regarding potential selection biases.
*Both parametric and non-parametric tests were used based on normality assumptions, but there is no explanation for why tests such as Mann-Whitney U and Kruskal-Wallis were chosen instead of transforming the data to meet normality assumptions. A brief justification would be helpful.
The use of stepwise regression is appropriate, but it should be clarified whether multicollinearity among predictors was checked (e.g., using the Variance Inflation Factor [VIF]) and whether the assumptions of linear regression were met (e.g., homoscedasticity, normality of residuals).

Validity of the findings

*Significant differences in paranoid ideation between amateur and semi-professional/professional referees are reported, but potential confounding factors (e.g., age, experience) that could explain these differences are not discussed. A more nuanced discussion would strengthen the interpretation.
*The finding that adaptive perfectionism does not significantly predict psychopathological symptoms contradicts some prior literature. The authors should discuss potential reasons for this discrepancy (e.g., measurement issues, sample characteristics).
*Although effect sizes are reported, their interpretation is sometimes inconsistent. For example, a Cohen’s d of 0.45 is described as a "medium effect size," but in some fields, this might be considered small. The authors should ensure their interpretations align with established conventions in sports psychology.
*It is not mentioned whether there were missing data and how they were handled. If missing data were present, the use of techniques such as multiple imputation or sensitivity analyses should be discussed.
*For the Kruskal-Wallis test, Bonferroni corrections were used for post hoc comparisons. While this is a conservative approach, it may increase the risk of Type II errors. Alternative methods (e.g., Holm-Bonferroni) could be considered and justified.
*The study makes a valuable contribution to the literature by highlighting the role of psychological inflexibility and perfectionism in the mental health of soccer referees. However, the authors could strengthen the discussion by situating their findings within broader theoretical frameworks (e.g., Acceptance and Commitment Therapy [ACT] for psychological inflexibility).
*The authors briefly mention the importance of interventions targeting psychological inflexibility and perfectionism but do not provide specific recommendations. Expanding on this point (e.g., suggesting ACT-based interventions or cognitive-behavioral strategies focused on perfectionism) would enhance the practical relevance of the study.
*The limitations section is thorough, but the authors should also address the potential impact of self-report bias (e.g., social desirability) and the cross-sectional design, which precludes causal inferences.
*The authors could propose specific longitudinal or experimental studies to address the limitations of the current design. For example, following referees over a season to assess changes in psychological inflexibility and perfectionism in response to stressors (e.g., high-stakes matches, injuries) would provide valuable insights.
Table 1:
*Clarify the meaning of "Z" and "U Mann-Whitney" in the header. While understandable for experts, explicitly stating that "Z" corresponds to the test statistic and "U" to the Mann-Whitney U value would improve clarity.
*Significant differences are reported for the "Paranoid Ideation" subscale (p = .029), but the effect size is not presented. Including the effect size (e.g., r or Cohen’s d) would help interpret the clinical relevance.
*In the t-test results, the notation "t (gl=154)" is confusing. It could be improved by using "t (df = 154)" to align with international standards.
*The table could benefit from an additional column presenting effect sizes (r for Mann-Whitney U and Cohen’s d for t-tests), as some significant differences may not be clinically meaningful.
Table 2:
*F and p values for ANOVA are presented without specifying whether post hoc analyses were conducted. Given the significant differences, it is suggested to include post hoc comparisons to identify between which groups the differences occur.
*The designation of "Low PI (a)", "Medium PI (b)", and "High PI (c)" may cause confusion. It would be clearer to use "Low Psychological Inflexibility," "Medium," and "High" in Spanish or to detail this in the table note.
*Adding the effect size (η² or partial η²) for the ANOVA results would help clarify the magnitude of the observed differences.
Table 3:
*Given that some correlations are moderate to high (e.g., AAQ-II and SA-45 TOTAL, r = .725), it is suggested to discuss in the text whether collinearity might be an issue, particularly in the regression analyses.
*Include shading or bolding to highlight moderate-to-high magnitude correlations (> .5) to facilitate visual interpretation.
Table 4:
*The models show marginal increases in R² between Model 2 and Model 3. It would be useful to include the F statistic for model comparisons and discuss the relevance of the improvement in explained variance.
*The β for MPS-PF-ADAP is not significant in Model 3 (p = .315), suggesting its inclusion may not be informative. A hierarchical regression analysis could be considered to better justify its inclusion.
Adding 95% confidence intervals for the β coefficients would improve the interpretation of the precision of the estimates.

Additional comments

Dear Authors,
First and foremost, I would like to acknowledge the work you have done and the effort invested in both the study and the preparation of the manuscript. In the sports context, the gap between research focused on athletes and that related to referees, particularly in soccer, is evident. This disparity has been noticeable across various areas concerning referees' health and well-being.
However, before considering the acceptance of your article, I believe it could benefit from some adjustments that would undoubtedly enhance the quality of the report and its scientific impact. With that in mind, I respectfully present my comments and suggestions for each section of the manuscript.

·

Basic reporting

Clear and unambiguous, professional English used throughout.
Literature references, sufficient field background/context provided.
Professional article structure, figures, tables. Raw data shared.
Self-contained with relevant results to hypotheses.

Experimental design

Research question well defined, relevant & meaningful. It is stated how research fills an identified knowledge gap
Rigorous investigation performed to a high technical & ethical standard.
Methods described with sufficient detail & information to replicate.

Validity of the findings

All underlying data have been provided; they are robust, statistically sound, & controlled.
Suggestions for future studies can be presented more clearly by adding a conclusion section.

Additional comments

There are very few studies on referees in the literature. This study conducted by the authors on referees is a very valuable study. It is very difficult to reach referee groups and conduct studies. In this context, I congratulate the authors for their work.

·

Basic reporting

All comments are attached in the PDF document

Experimental design

All comments are attached in the PDF document

Validity of the findings

All comments are attached in the PDF document

Additional comments

Thanks for the opportunity to review this paper. The paper evaluates the relationship between
psychological inflexibility, perfectionism, and psychopathological symptomatology in amateur and semi-professional soccer referees. The study is well-designed, follows a generally accepted methodology, and presents novel findings. It is well-written, except for the discussion section, which requires major revisions. Specifically, the discussion section largely repeats the results, lacks interpretation, and does not suggest practical implementations. I believe the authors can address these issues. Please refer to the specific comments in the PDF document.
Regards

---

## Round 0.2 · accepted · Accept

I believe the authors have addressed all issues raised by the reviewers and the manuscript now meets the standard for publication.

·

Basic reporting

Clear and unambiguous, professional English used throughout. Literature references, sufficient field background/context provided. Professional article structure, figures, tables. Raw data shared.

Experimental design

Original primary research within Aims and Scope of the journal. Research question well defined, relevant & meaningful. It is stated how research fills an identified knowledge gap. Methods described with sufficient detail & information to replicate.

Validity of the findings

Impact and novelty not assessed. Meaningful replication encouraged where rationale & benefit to literature is clearly stated.All underlying data have been provided; they are robust, statistically sound, & controlled

·

Basic reporting

This study is of reasonable quality but requires substantial revision before it can be considered for publication. The main limitations include: (1) limited discussion of the findings in relation to the existing literature; (2) overly long and unfocused sentences in the Introduction and Discussion sections that do not clearly support the study’s aims; (3) repetition of content from the Tables in the Results section; and (4) a lack of discussion on practical implications. The authors can find my comments in the PDF document

Experimental design

All were stated in the document

Validity of the findings

All were stated in the document

Additional comments

All were stated in the document